# A Stable, Fast, and Fully Automatic Learning Algorithm for Predictive Coding Networks

**Tommaso Salvatori**[1,5,*]**, Yuhang Song**[2,6,*,†]**,Yordan Yordanov**[3]**, Beren Millidge**[2]
**Cornelius Emde**[3]**, Zhenghua Xu**[4]**, Lei Sha**[3]**, Rafal Bogacz**[2]**, Thomas Lukasiewicz**[5,3]
[1] VERSES AI Research Lab, Los Angeles, California, 90016, USA
[2] MRC Brain Network Dynamics Unit, University of Oxford, UK
[3]Department of Computer Science, University of Oxford, UK
[4] School of Health Sciences and Biomedical Engineering, Hebei University of Technology, China
[5] Institute of Logic and Computation, Vienna University of Technology, Austria
[6] Fractile Ltd, London, UK
`tommaso.salvatori@verses.ai, thomas.lukasiewicz@tuwien.ac.at`
 `{yuhang.song,beren.millidge,rafal.bogacz}@ndcn.ox.ac.uk`
 `{yordan.yordanov,lei.sha,cornelius.emde}@cs.ox.ac.uk`

## Abstract

Predictive coding networks are neuroscience-inspired models with roots in both Bayesian statistics and neuroscience. Training such models, however, is quite inefficient and unstable. In this work, we show how by simply changing the temporal scheduling of the update rule for the synaptic weights leads to an algorithm that is much more efficient and stable than the original one, and has theoretical guarantees in terms of convergence. The proposed algorithm, which we call incremental predictive coding (iPC), is also more biologically plausible than the original one, as it is fully automatic. In an extensive set of experiments, we show that iPC constantly performs better than the original formulation on a large number of benchmarks for image classification, as well as for the training of both conditional and masked language models, in terms of test accuracy, efficiency, and convergence with respect to a large set of hyperparameters.

## 1 Introduction

In recent years, deep learning has reached and surpassed human-level performance in a multitude of tasks, such as game playing (Silver et al., 2017; 2016; Bakhtin et al., 2022), image recognition (Krizhevsky et al., 2012; He et al., 2016), natural language processing (Chen et al., 2020), and image generation (Ramesh et al., 2022; Saharia et al., 2022). These successes are achieved entirely using deep artificial neural networks trained via *backpropagation* (*BP*), which is a learning algorithm that is often criticized for its biological implausibilities (Grossberg, 1987; Crick, 1989; Abdelghani et al., 2008; Lillicrap et al., 2016; Roelfsema & Holtmaat, 2018; Whittington & Bogacz, 2019), such as lacking local plasticity and autonomy. In fact, backpropagation requires a global control signal to trigger computations, since gradients must be sequentially computed backwards through the computation graph. The biological plausibility of a specific algorithm is not a niche interest of theoretical neuroscience, but it is of vital importance when it comes to implementations on low-energy analog/neuromorphic chips: *parallelization*, *locality*, and *automation* are key to building efficient models that can be trained end-to-end on non-von-Neumann machines, such as analog chips (Kendall et al., 2020). To this end, multiple works are highlighting the need of fueling fundamental research in computational neuroscience to find algorithms and methods that can solve the aforementioned problems (Zador et al., 2022; Friston et al., 2022). A promising learning algorithm in this regard, which has most of the above properties, is predictive coding (PC).

PC is an influential theory of information processing in the brain (Mumford, 1992; Friston, 2005), where learning happens by minimizing the prediction error of every neuron. PC can be shown to

---
[*]Equal contribution.
[†]Corresponding author.

approximate BP in layered networks (Whittington & Bogacz, 2017), as well as on any other model (Millidge et al., 2020a), and can exactly replicate its weight update if some external control is added (Salvatori et al., 2022b). Also, the differences with BP are interesting, as PC allows for a much more flexible training and testing (Salvatori et al., 2022a), has a rich mathematical formulation (Friston, 2005; Millidge et al., 2022a), and is an energy-based model (Bogacz, 2017). Simply put, PC is based on the assumption that brains implement an internal generative model of the world, needed to predict incoming stimuli (or data) (Friston et al., 2006; Friston, 2010; Friston et al., 2016). When presented with a stimulus that differs from the prediction, learning happens by updating internal neural activities and synapses to minimize the *prediction error*. In computational models, this is done via a minimization of the *variational free energy*, in this case a function of the total error of the generative model. This minimization happens in two steps: first, internal neural activities are updated in parallel until convergence; then, synaptic weights are updated to further minimize the same energy function. This brings us to the second peculiarity of PC, which is its solid statistical formulation, developed much before its links to neuroscience (Elias, 1955). The message passing scheme of predictive coding is, in fact, an efficient way of inverting a hierarchical Gaussian generative model by approximating an evidence lower bound using Laplace and mean field approximations (Friston, 2005; Friston et al., 2008).

When applying this inversion schema to large-scale neural network training, we encounter three limitations: first, an external control signal is needed to switch from the step that updates the neural activities, and the update of the synaptic weights; second, the update of the neural activities is slow, as it can require dozens of iterations to converge; third, convergence is uncertain and highly dependent on the choice of hyperparameters. Consequently, researchers working in the field have always struggled with the slow training of predictive coding models, as well as the extensive hyperparameter tuning needed to reach the optimal performance. Here, we address these problems by considering a variant of PC where the update of both the value nodes and the parameters are performed in parallel, similarly to how it is done in (Ernoult et al., 2020). This algorithm is provably faster, does not require a control signal to switch between the two steps, performs empirically much better, has solid convergence guarantees, and is more robust to changes in hyperparameters. We call this training algorithm *incremental predictive coding* (iPC). Our contributions are briefly as follows:

1. We first present the update rule of iPC, and discuss the implications of this change in terms of autonomy, and its differences and similarities with PC and BP. Then, we show its convergence guarantees by deriving the same equations from the variational free energy of a hierarchical generative model using the incremental expectation-maximization approach (iEM): it has in fact been proven that iEM converges to a minimum of the loss function (Neal & Hinton, 1998; Karimi et al., 2019), and hence this result naturally extends to iPC.

2. We empirically compare the efficiency of PC and iPC on generation and classification tasks. In both cases, iPC is by far more efficient than the original counterpart, as well as reaching a better performance by converging to better local minima. We also compare its efficiency with that of BP in the special case of full batch training.

3. We then test our method on image classification benchmarks as well as conditional and masked language models, showing that iPC performs better than PC, and that the more complex the task, the larger the gap in performance. We then explore metrics that go beyond standard test accuracies, and show that the best performing models trained with PC have well-calibrated outputs, and that iPC is more parameter-efficient than BP.

## 2 PRELIMINARIES

In this section, we introduce the original formulation of predictive coding as a generative model proposed by Rao and Ballard (1999). Consider a generative model $g \colon \mathbb{R}^d \times \mathbb{R}^D \longrightarrow \mathbb{R}^o$, where $x \in \mathbb{R}^d$ is a vector of latent variables, called *causes*, $y \in \mathbb{R}^o$ is the generated vector, and $\theta \in \mathbb{R}^D$ is a set of parameters. We are interested in the following inverse problem: given a vector $y$ and a generative model $g$, we need the parameters $\theta$ that maximize the marginal likelihood

$$p(y, \theta) = \int_x p(y \mid x, \theta) p(x, \theta) dx. \tag{1}$$

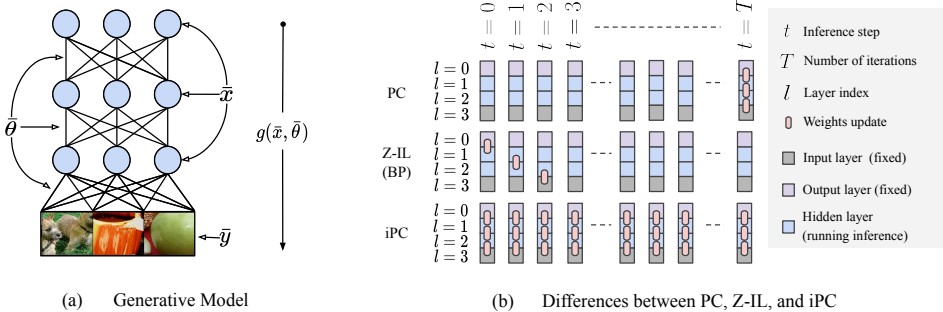

(a)    Generative Model               (b)    Differences between PC, Z-IL, and iPC

Figure 1: (a) An example of a hierarchical Gaussian generative model with three layers. (b) Comparison of the temporal training dynamics of PC, Z-IL, and iPC, where Z-IL is a variant of PC that is equivalent to BP, originally introduced in (Song et al., 2020). We assume that we train the networks on a dataset for supervised learning for a period of time $T$. Here, $t$ is the time axis during inference, which always starts at $t = 0$. The squares represent nodes in one layer, and pink rounded rectangles indicate when the connection weights are modified: PC (1st row) first conducts inference on the hidden layers, according to Eq. (6), until convergence, and then it updates the weights via Eq. (7). Z-IL (2nd row) only updates the weights at specific inference moments depending on which layer the weights belong to. To conclude, iPC updates the weights at every time step $t$, while performing inference in parallel.

Here, the first term inside the integral is the likelihood of the data given the causes, and the second is a prior distribution over the causes. Solving the above problem is intractably expensive. Hence, we need an algorithm that is divided into two phases: *inference*, where we infer the best causes $x$, given both $\theta$ and $y$, and *learning*, where we update the parameters $\theta$ based on the newly computed causes. This algorithm is *expectation-maximization* (EM) (Dempster et al., 1977). The first step, which we call inference or E-step, computes $p(x \mid y, \theta)$, which is the posterior distribution of the causes given a generated vector $y$. Computing the posterior is, however, intractable (Friston, 2003). To this end, we approximate the intractable posterior with a tractable probability distribution $q(x, \theta)$. To make the approximation as good as possible, we want to minimize the KL-divergence between the two probability distributions. Summarizing, to solve our learning problem, we need to (i) minimize a KL-divergence, and (ii) maximize a likelihood. We do it by defining the following energy function, also known as *variational free energy*:

$$F(x, y, \theta) = KL(q(x, \theta)\|p(x \mid y, \theta)) - ln(p(y, \theta)), \tag{2}$$

where we have used the log-likelihood. This function is minimized by multiple iterations of the EM algorithm:

$$\begin{cases} \text{Inference (E-step): } x^* = argmax_x F(x, y, \theta), \\ \text{Learning (M-step): } \theta^* = argmax_\theta F(x, y, \theta). \end{cases} \tag{3}$$

## 2.1 Predictive Coding

So far, we have only presented the general problem. To actually derive proper equations for learning causes and update the parameters, and use them to train neural architectures, we need to specify the generative function $g(x, \theta)$. Following the general literature (Rao & Ballard, 1999; Friston, 2005), we define the generative model as a hierarchical Gaussian generative model, where the causes $x$ and parameters $\theta$ are defined by a concatenation of the causes and weight matrices of all the layers, i.e., $x = (x^{(0)}, \ldots, x^{(L)})$ and $\theta = (\theta^{(0)}, \ldots, \theta^{(L-1)})$. Hence, we have a multilayer generative model, where layer 0 is the one corresponding to the generated image $y$, and layer $L$ is the highest in the hierarchy. The marginal probability of the causes is as follows:

$$p(x^{(0)}, \ldots, x^{(L)}) = \prod_l^L \mathcal{N}(\mu^{(l)}, \Sigma^{(l)}), \tag{4}$$

where $\mu^{(l)}$ is the *prediction* of layer $l$ according to the layer above, given by $\mu^{(l)} = \theta^{(l)} \cdot f(x^{(l+1)})$, with $f$ being a non-linear function and $\mu^{(L)} = x^{(L)}$. For simplicity, from now on, we consider

---

**Algorithm 1** Learning a dataset $\mathcal{D} = \{y_i\}$ with iPC.

---

1: **Require:** For every $i$, $x_i^{(0)}$ is fixed to $y_i$,
2: **for** $t = 0$ to $T$ **do**
3:     For every $i$ and $l$, update $x_i^{(l)}$ to minimize $F$ via Eq. (6)
4:     For every $l$, update each $\theta^{(l)}$ to minimize $F$ via Eq. (7)
5: **end for**

---

Gaussians with identity variance, i.e., $\Sigma^{(l)} = \mathbb{1}$ for every layer $l$. With the above assumptions, the free energy becomes

$$F = \sum_l \|x^{(l)} - \mu^{(l)}\|^2. \tag{5}$$

For a detailed formulation on how this energy function is derived from the variational free energy of Eq. (2), we refer to (Friston, 2005; Bogacz, 2017; Buckley et al., 2017; Millidge et al., 2021). Note that this energy function is equivalent to the one proposed in the original formulation of PC (Rao & Ballard, 1999). A key aspect of this model is that both inference and learning are achieved by optimizing the same energy function, which aims to minimize the prediction error of the network. The prediction error of every layer is given by the difference between its real value $x^{(l)}$ and its prediction $\mu^{(l)}$. We denote the prediction error by $\varepsilon^{(l)} = x^{(l)} - \mu^{(l)}$. Thus, the problem of learning the parameters that maximize the marginal likelihood given a data point $y$ reduces to an alternation of inference and weight update. During both phases, the values of the last layer are fixed to the data point, i.e., $x^{(0)} = y$ for each $t \leq T$.

**Inference:** During this phase, which corresponds to the E-step, the weight parameters $\theta^{(l)}$ are fixed, while the values $x^{(l)}$ are continuously updated via gradient descent:

$$\Delta x^{(l)} = \gamma \cdot (-\varepsilon^{(l)} + f'(x^{(l)}) * \theta^{(l-1)\ \mathsf{T}} \cdot \varepsilon^{(l-1)}), \tag{6}$$

where $*$ denotes element-wise multiplication, and $l > 0$. This process either runs until convergence, or for a fixed number of iterations $T$.

**Learning:** During this phase, which corresponds to the M-step, the values $x$ are fixed, and the weights are updated once via gradient descent according to the following equation:

$$\Delta\theta^{(l)} = -\alpha \cdot \partial F / \partial\theta^{(l)} = \alpha \cdot f(x^{(l+1)})\varepsilon^{(l)}. \tag{7}$$

Note that the above algorithm is not limited to generative tasks, but can also be used to solve supervised learning problems (Whittington & Bogacz, 2017). Assume that a data point $y_{in}$ with label $y_{out}$ is provided. In this case, we treat the label as the vector $y$ that we need to generate, and the data point as the prior on $x^{(L)}$. The inference and learning phases are identical, with the only difference that now we have two vectors fixed during the whole duration of the process: $x^{(0)} = y_{out}$ and $x^{(L)} = y_{in}$. While this algorithm is able to obtain good results on small image image classification tasks, it is much slower than BP due to the large number of inference steps $T$ needed to let the causes $x$ converge. In what follows, we propose an algorithm that addresses this limitation.

We have defined PC on hierarchical Gaussian models, as different probability distributions would result in update rules that do not minimize prediction errors (Salvatori et al., 2023). However, the applicability of our algorithm can be easily generalized to different probability distributions (Pinchetti et al., 2022), as we also show in Section 5.

## 3   INCREMENTAL PREDICTIVE CODING

What makes PC much slower than BP is its inference phase, which requires multiple iterations to converge. In this section, we address this limitation by proposing *incremental* PC, a variation of the original algorithm where the inference and learning phases (Eqs. (6) and (7)) are simultaneously performed at every time step $t$. This variation largely improves on the original formulation of PC in terms of both efficiency and performance, is fully automatic, and comes with theoretical guarantees given by the theory of variational inference. The pseudocode of iPC is provided in Algorithm 1, while its dynamics is illustrated in Fig. 1(b).

**Connections to BP:** PC in general shares multiple similarities with BP in supervised learning tasks: when the output error is small, the parameter update of PC is an approximation of that of BP (Millidge et al., 2020a); when controlling which parameters have to be updated at which time step, it is possible to define a variation of PC, called *zero-divergence inference learning* (Z-IL) in the literature, whose updates are equivalent to those of BP (Song et al., 2020). In detail, to make PC perform exactly the same weight updates of BP, every weight matrix $\theta^l$ must be updated only at $t = l$, which corresponds to its position in the hierarchy. That is, as soon as the output error reaches a specific layer. This is different from the standard formulation of PC, which updates the parameters only when the energy representing the total error has converged. Unlike PC, iPC updates the parameters at every time step $t$. Intuitively, it can hence be seen as a "continuous shift" between Z-IL (and hence BP) and PC. A graphical representation of the differences of all three algorithms is given in Fig. 1 (right), with the pseudo-codes provided in the first section of the supplementary material.

**Autonomy:** Both PC and Z-IL lack full autonomy, as an external control signal is always needed to switch between inference and learning: PC waits for the inference to converge (or for $T$ iterations), while Z-IL updates the weights of specific layers at specific inference moments $t = l$. BP is considered to be less autonomous than PC and Z-IL: a control signal is required to forward signals as well as backward errors, and additional places to store the backward errors are required. All these drawbacks are removed in iPC, where the only control signal needed is the switching among different batches. In a full-batch training regime, however, iPC is able to learn a dataset without the control signals required by the other algorithms: given a dataset $\mathcal{D}$, iPC runs inference and weight updates simultaneously until the energy $F$ is minimized. As soon as the energy minimization has converged, training ends.

**Incremental EM:** iPC can also be derived from the variational free energy of Eq. (2), and minimize it using a variation of the EM, precisely developed to address the lack of efficiency of the original algorithm when dealing with multiple data points at the same time, a scenario that is almost always present in standard machine learning. This alternative form, which we now present, is called *incremental* EM (iEM) (Neal & Hinton, 1998). Let $\mathcal{D} = \{y_i\}_{i<N}$ be a dataset of cardinality $N$, and $g(x, \theta)$ be a generative model. Our goal is now to minimize the global marginal likelihood, defined on the whole dataset, i.e.,

$$p(\mathcal{D}, \theta) = \sum\nolimits_i p(y_i, \theta). \tag{8}$$

The same reasoning also applies to the global variational free energy, which is the sum of the free energies of every single data point. In this case, the iEM algorithm performs the E-step and M-step in parallel, with no external control needed to switch between the two phases. That is, both the values $x$ and the parameters $\theta$ are updated simultaneously at every time step $t$, until convergence, on all the points of the dataset. No explicit forward and backward passes are necessary, as each layer is updated in parallel. This also comes with strong theoretical guarantees, as it has been formally proven that minimizing a free-energy function such as ours (i.e., equivalent to the sum of independent free-energy functions) using iEM, also finds a minimum of the global marginal likelihood of Eq. (8) (Neal & Hinton, 1998; Karimi et al., 2019). We actually provide empirical evidence that the model converges to better minima using iPC rather than the original formulation of PC in Fig. 2 and Table 1. The pseudocode of iPC is given in Alg. 1.

## 3.1 EFFICIENCY

In this section, we analyze the efficiency of iPC relative to both the original formulation of PC and BP. We only provide partial evidence of the increased efficiency against BP, as standard deep learning frameworks, such as PyTorch, do not allow to parallelize operations in different layers.

**Comparison with PC:** We now show how iPC is more efficient than the original formulation. To do that, we have trained multiple models with iPC and PC on different tasks and datasets. First, we have trained a generative model with $4$ layers and $256$ hidden neurons on a subset of $100$ images of the Tiny ImageNet and CIFAR10 datasets, exactly as in (Salvatori et al., 2021). A plot with the energies as a function of the number of iterations is in Fig. 2 (left and centre). In both cases, the network trained with iPC converges much faster than the networks trained with PC with different values of $T$. Many more plots with different parameterizations are given in the supplementary material.

To show that the above results hold in different setups as well, we have trained a classifier with $4$ layers on a subset of $250$ images of the FashionMNIST dataset, following the framework proposed in

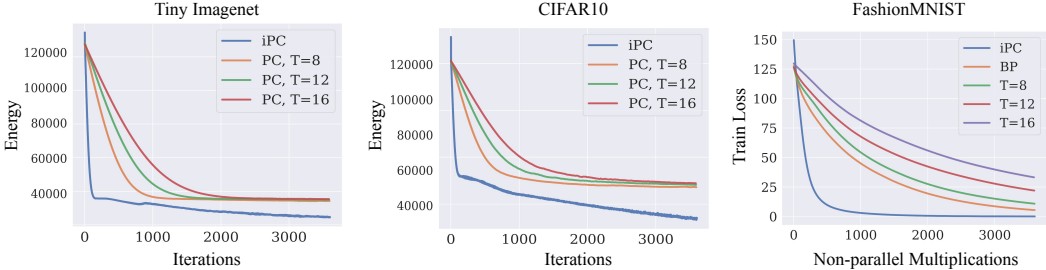

Figure 2: Left and centre: Decrease of the energy of generative models as a function of the number of iterations performed from the beginning of the training process. Right: Training loss of different classifiers in a full-batch training regime as a function of the number of non-parallel matrix multiplications performed from the beginning of the training process.

Table 1: Test accuracy of BP, PC, and iPC on different architectures trained with different datasets. *Data augmentation was used here.

|  | BP | PC | iPC |
|---|---|---|---|
| MLP on MNIST | $98.26\% \pm 0.12\%$ | $\mathbf{98.55}\% \pm 0.14\%$ | $98.54\% \pm 0.86\%$ |
| MLP on FashionMNIST | $88.54\% \pm 0.64\%$ | $85.12\% \pm 0.75\%$ | $\mathbf{89.13}\% \pm 0.86\%$ |
| CNN on SVHN | $95.35\% \pm 1.53\%$ | $94.53\% \pm 1.54\%$ | $\mathbf{96.45}\% \pm 1.04\%$ |
| CNN on CIFAR-10 | $69.34\% \pm 0.54\%$ | $70.84\% \pm 0.64\%$ | $\mathbf{72.54}\% \pm 0.93\%$ |
| AlexNet on CIFAR-10 | $\mathbf{75.64}\% \pm 0.64\%$ | $64.63\% \pm 1.55\%$ | $72.42\% \pm 0.53\%$ |
| AlexNet on CIFAR-10* | $\mathbf{83.12}\% \pm 0.97\%$ | $71.99\% \pm 2.43\%$ | $80.11\% \pm 0.44\%$ |

(Whittington & Bogacz, 2017), and studied the training loss. As it is possible to train an equivalent model using BP, we have done it using the same set-up and learning rate, and included it in the plot. This, however, prevents us from using the number of iterations as an efficiency measure, as one iteration of BP is more complex than one iteration of PC, and are hence not comparable. As a metric, we have hence used the number of non-parallel matrix multiplications needed to perform a weight update. This is a fair metric, as matrix multiplications are by far the most expensive operation performed when training neural networks, and the ones with largest impact on the training speed. Single iterations of PC and iPC have the same speed, and consist of 2 non-parallel matrix multiplications. One epoch of BP consists of $2L$ non-parallel matrix multiplications. The results are given in Fig. 2 (right). In all cases, iPC converges much faster than all the other methods. In the supplementary material, we provide other plots obtained with different datasets, models, and parameterizations, as well as a study on how the test error decreases during training.

**Comparison with BP:** While the main goal of this work is simply to overcome the core limitation of original PC (namely, the slow inference phase), there is one scenario where iPC is potentially more efficient than BP, which is full batch training. Particularly, we first prove this formally using the number of non-parallel matrix multiplications needed to perform a weight update as a metric. To complete one weight update, iPC requires two sets of non-parallel multiplications: the first uses the values and weight parameters of every layer to compute the prediction of the layer below; the second uses the error and transpose of the weights to propagate the error back to the layer above, needed to update the values. BP, on the other hand, requires $2L$ sets of non-parallel multiplications for a complete update of the parameters: $L$ for a forward pass, and $L$ for a backward one. These operations cannot be parallelized. More formally, we prove a theorem that holds when training on the whole dataset $\mathcal{D}$ in a full-batch regime. For details about the proof, an extensive discussion about time complexity of BP, PC, and iPC, we refer to the supplementary material.

**Theorem 3.1.** *Let $M$ and $M'$ be two equivalent networks with $L$ layers trained on the same dataset. Let $M$ be trained using BP, and $M'$ be trained using iPC. Then, the time complexity needed to perform one full update of the weights is $\mathcal{O}(1)$ for iPC and $\mathcal{O}(L)$ for BP.*

Table 2: Change of final accuracy when increasing the width.

| $C$ | 1 | 2 | 3 | 4 | 5 | 6 | 7 | 8 | 10 | 15 | 20 |
|---|---|---|---|---|---|---|---|---|---|---|---|
| BP | 67.92 | 71.23 | 71.65 | 72.64 | 73.35 | 73.71 | 74.19 | 74.51 | 74.62 | 75.08 | 75.51 |
| iPC | **70.61** | **74.12** | **74.91** | **75.88** | **76.61** | **77.04** | **77.48** | **77.41** | **76.51** | **76.55** | **76.12** |

## 4 CLASSIFICATION EXPERIMENTS

We now demonstrate that iPC shows a similar level of generalization quality compared to BP. We test the performance of iPC on different benchmarks. Since we focus on generalization quality in this section, all methods are run until convergence, and we have used early stopping to pick the best performing model. These experiments were performed using multi-batch training. In this case, we lose our advantage in efficiency over BP, as we need to recompute the error every time a new batch is presented. However, the proposed algorithm is still much faster than the original formulation of PC, and yields a better classification performance.

**Setup of experiments:** We investigate image classification benchmarks using PC, iPC, and BP. We first trained a fully connected network with 2 hidden layers and 64 hidden neurons per layer on the MNIST dataset (LeCun & Cortes, 2010). Then, we trained a mid-size CNN with three convolutional layers with $64 - 128 - 64$ kernels followed by two fully connected layers on FashionMNIST, the Street View House Number (SVHN) dataset (Netzer et al., 2011), and CIFAR10 (Krizhevsky et al., 2012). Finally, we trained AlexNet (Krizhevsky et al., 2012), a large-scale CNN, on CIFAR10. To make sure that our results are not the consequence of a specific choice of hyperparameters, we performed a comprehensive grid-search on hyperparameters (more details in the supplementary material), and reported the highest test accuracy obtained. We have also carefully checked whether the energy/loss of every model had converged, and this was indeed the case. Hence, the worse performance of PC on AlexNet is probably due to scaling properties of PC, rather than a non-converged network. This is a problem that we have not experienced using iPC, able to well scale to larger architectures.

**Convergence:** In the experiments on AlexNet, under all the hyperparameter combinations tested, iPC only failed to converge when the learning rate of the weights was the largest (0.01). In total, it converged 88 times out of 96 combinations of hyperparameters. This is not the case for PC, which converged in only 26 combinations of hyperparameters out of 96 (We consider a model to be converged, if the difference between its best test accuracy, and the best test accuracy reached over the whole hyperparameter search, is less than 10%).

**Results:** In Table 1, iPC constantly outperforms PC, besides in the simplest framework (MNIST on a small MLP), where PC has a tiny margin of 0.01%. But PC fails to scale to more complex problems, where it gets outperformed by all the other training methods. The performance of iPC, on the other hand, is stable under changes in size, architecture, and dataset, and is comparable to the one of BP.

**Change of width:** To investigate how iPC behaves when adding max-pooling layers and increasing the width, we trained a CNN with three convolutional layers $(8, 16, 8)$ and maxpools, followed by a fully connected layer (128 hidden neurons) on CIFAR10. We have also replicated the experiment by increasing the width of the network by multiplying every hidden dimension by a constant $C$ (e.g., $C = 3$ means a network with 3 convolutional layers $(24, 48, 24)$, each followed by a maxpool, and a fully connected one (384 hidden neurons)). The results in Table 2 show that iPC (i) outperforms BP under each parametrization, (ii) needs less parameters to obtain good results, but (iii) sees its performance decrease, once it has reached a specific parametrization. This is in contrast to BP, which is able to generalize well even when extremely overparametrized. This suggests that iPC is more efficient than BP in terms of the number of parameters, but that finding the best parameters for iPC may need some extra tuning.

### 4.1 ROBUSTNESS AND CALIBRATION

Robustness and uncertainty quantification in deep learning have become a topic of increasing interest in recent years. Recently, it has been noted that treating classifiers as generative models benefits the

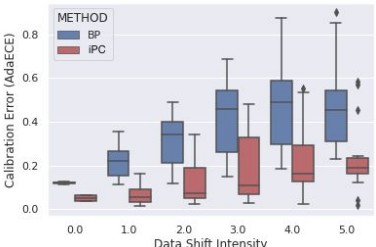 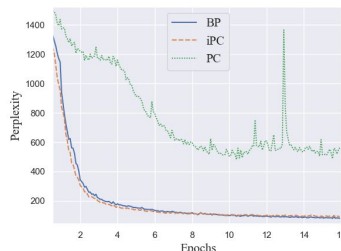

Figure 3: Left: Robustness of BP and iPC under distribution shift (AlexNet on CIFAR10 under five different intensities of the corruptions rotation, Gaussian blur, Gaussian noise, hue, brightness, and contrast). iPC maintains model calibration significantly better than BP under distribution shift. Right: Dev perplexity during training of the best performing masked language models.

robustness of the model (Grathwohl et al., 2019). The same result is obtained by adding layers, or message passing schemes, that simulate the visual cortex of primates (Dapello et al., 2020; Choksi et al., 2021). Specifically about PC, it has been shown that the training procedure is more stable, as it replicates explicit gradient descent schemas (Alonso et al., 2022), and that it learns more robust representations (Song et al., 2022; Byiringiro et al., 2022). We now empirically show that this also extends to iPC by comparing its robustness and calibration capabilities with the ones of BP. Calibration describes the degree to which predicted logits matches the empirical distribution of observations given the prediction confidence. One may use the output of a calibrated model to quantify the uncertainty in its predictions and interpret it as probability—not just model confidence. Let $\hat{P}$ be our random prediction vector indicating the confidence that the prediction $\hat{Y}$ is correct. We say that $\hat{P}$ is well-calibrated, if the model confidence matches the model performance, i.e., $\mathbb{P}(\hat{Y} = Y | \hat{P} = p) = p$ (Guo et al., 2017). We measure the deviation from calibration using the adaptive expected calibration error (AdaECE), which estimates $\mathbb{E}[|\mathbb{P}(\hat{Y} = Y | \hat{P} = p) - p|]$ (Nguyen & O'Connor, 2015). In recent years, it has become well-known that neural networks trained with BP tend to be overconfident in their predictions (Guo et al., 2017) and that miscalibration increases dramatically under distribution shift (Ovadia et al., 2019).

**Results:** Our results are shown in Fig. 3. The boxplot indicates the distributions of calibration error over various forms of data corruption with equal levels of intensity, which differ strongly between iPC and BP: The iPC-trained model yields better calibrated outputs and is able to much better signal its confidence. This is essential for using the model output as indication of uncertainty. On in-distribution data, we observe that iPC yields an average calibration error of $0.05$, whereas BP yields $0.12$. Moreover, we observe that the increase in calibration error is a lot weaker for iPC: The median calibration error of the iPC model is lower across all levels of shift intensities compared to that of BP for the mildest corruption. Furthermore, iPC displays better calibration up to level 3 shifts than BP does on in-distribution data. This has potentially a strong impact of applying either method in safety-critical applications.

## 5 LANGUAGE MODEL EXPERIMENTS

A recent work has shown that it is possible to introduce a small modification to the training algorithm of PC to improve its performance on small language models (LMs) (Pinchetti et al., 2022). Here, we test the performance of PC, iPC, and BP on BERT, a popular encoder-only transformer language model architecture (Devlin et al., 2019). This model is trained to reconstruct randomly masked tokens from the input. To improve the range of our study, we also train a conditional version of BERT, where we add a triangular mask to the attention mechanism, so that the model generates each token only based on the previous tokens in the text. This creates a decoder-only language model with a similar architecture to GPT (Radford et al., 2018).

**Setup:** The training and dev datasets are generated by randomly sampling $200,000$ and $10,000$ instances, respectively, from the One Billion Word Benchmark (Chelba et al., 2013). The test dataset is the original test dataset of the 1B Word Benchmark. For both models, we use two transformer blocks with one head and a hidden size of $128$. The vocabulary is obtained via byte-pair-encoding with $8001$ tokens, generated via the SentencePiece tokenizer (Kudo & Richardson, 2018). After

the best hyperparameters are selected, we run each method on 9 additional seeds for a total of 10 seeds. This allows us to compare the expected perplexity (ppl) performance and see the performance variation across seeds. We also use a convergence threshold to discard those models that do not converge. For iPC and BP, we define the convergence threshold at 200 test perplexity, and for PC, we define it at 800. For complete per-seed results, as well as all the details needed to reproduce the results, we refer to the supplementary material.

Table 3: Perplexity of the three compared methods on 10 random seeds, for both the masked and conditional language models, with the number of converged models.

| Model | BP | | PC | | iPC | |
|---|---|---|---|---|---|---|
| | Perplexity | #Conv | Perplexity | #Conv | Perplexity | #Conv |
| Masked LM | $120.02 \pm 13.19$ | 7 | $523.08 \pm 12.3$ | 3 | $\mathbf{106.19} \pm 10.54$ | **10** |
| Cond. LM | $\mathbf{113.32} \pm 0.36$ | **10** | $206.34 \pm 6.46$ | **10** | $142.54 \pm 4.23$ | 7 |

**Results:** Our experiments show that iPC significantly outperforms PC in both masked and conditional language models. For masked LMs, iPC also exhibits a much better convergence, with all 10 seeds converging, whereas PC has only 3 converging seeds. The poor performance of PC is due to its poor training stability, as evident by Fig. 3 (right), where we can also see that the training curves of iPC and BP are similar. In fact, iPC performs similarly to BP in terms of test perplexity, with iPC performing better than BP on masked LMs with 106 vs. 120 ppl (where all 10 runs converged), and worse on conditional LMs with 113 vs. 143 ppl (where 3 runs have not converged). The results are reported in Table 3. We can then conclude that the experiments performed on language models showed that iPC is significantly better than PC in terms of both performance and stability, obtaining results that are comparable to those of BP.

## 6 RELATED WORKS

Neuroscience-inspired algorithms have recently gained the attention of the machine learning community, and multiple works have used PC to tackle machine learning problems, from generation tasks (Ororbia & Kifer, 2020), to image classification on complex datasets such as ImageNet (He et al., 2016), associative memories (Salvatori et al., 2021; Tang et al., 2023), continual learning (Ororbia et al., 2020), and NLP (Pinchetti et al., 2022). In terms of potential implementations on neuromorphic chips, there are multiple lines of work that are parallel to PC, such as *local representation alignment* (Ororbia II et al., 2017; Ororbia & Mali, 2019), *equilibrium propagation* (Scellier & Bengio, 2017), *feedback alignment* (Lillicrap et al., 2016), and *SoftHebb* (Journé et al., 2022). Theoretical works, on the other hand, have studied the similarities between PC, backpropagation, and the aforementioned algorithms (Millidge et al., 2022b;c).

## 7 DISCUSSION

Researchers working in the field of predictive coding have certainly experienced the slow and unstable training of predictive coding networks. In this paper, we have proposed a variation of PC that enables all the computations to be executed *simultaneously*, *locally*, and *autonomously*, and has theoretical convergence guarantees in non-asymptotic time (Karimi et al., 2019). This allows a solid gain in efficiency compared to the original formulation of PC, as shown with extensive experiments, as well as improved performance and robustness in all the considered tasks. Many other works that speed up training algorithms and converge to better minima, such as ADAM optimization (Kingma & Ba, 2014), have had a huge impact in the communities that they were proposed in, while also being simple on the surface. Similarly, we forsee that many researchers working in PC can now use the proposed update rule, which comes with no apparent drawbacks with respect to the original one. The fact that it empirically converges to better minima also allows PC to reach a performance comparable to those of BP on complex tasks, such as image classification in convolutional models, or language generation in transformer models.

## 8 AKNOWLEDGEMENTS

Beren Millidge and Rafal Bogacz were supported by BBSRC grant BB/S006338/1. Rafal Bogacz was supported by MRC grant MC_UU_00003/1. This work was also supported by the AXA Research Fund, by the Alan Turing Institute under the EPSRC grant EP/N510129/1, and by the EPSRC grant EP/R013667/1. We also acknowledge the use of the EPSRC-funded Tier 2 facility JADE (EP/P020275/1) and GPU computing support by Scan Computers International Ltd. C. Emde is supported by the EPSRC Centre for Doctoral Training in Health Data Science (EP/S02428X/1) and Cancer Research UK (CRUK).

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

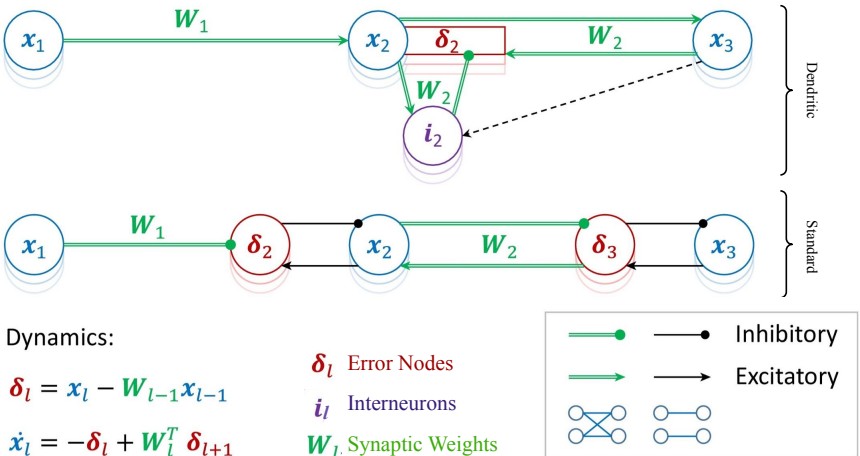

Figure 4: Standard and dendritic neural implementation of predictive coding. The dendritic implementation makes use of interneurons $i_l = W_l x_l$ (according to the notation used in the figure). Both implementations have the same equations for all the updates, and are thus equivalent; however, dendrites allow a neural implementation that does not take error nodes into account, improving the biological plausibility of the model. Figure taken and adapted from (Whittington & Bogacz, 2019).

## A A DISCUSSION ON BIOLOGICAL PLAUSIBILITY

In this section, we discuss the biological plausibility of the proposed algorithm. In the literature, there is often a disagreement on whether a specific algorithm is biologically plausible or not. Generally, it is assumed that an algorithm is biologically plausible when it satisfies a list of properties that are also satisfied in the brain. Different works consider different properties. In our case, we consider as list of minimal properties that include local computations and a lack of a global control signals to trigger the operations. Normally, predictive coding networks take error nodes into account, often considered implausible from the biological perspective (Sacramento et al., 2018). Even so, the biological plausibility of our model is not affected by this: it is in fact possible to map PC on a different neural architecture, in which errors are encoded in apical dendrites rather than separate neurons (Sacramento et al., 2018; Whittington & Bogacz, 2019). Graphical representations of the differences between the two implementations can be found in Fig. 4, taken (and adapted) from (Whittington & Bogacz, 2019). Furthermore, our formulation is more plausible than the original formulation of PC, as it is able to learn without the need of external control signals that trigger the weight update.

**Weight Transport:** While predictive coding is designed to model information processing in various brain regions, the current formulation still exhibits certain biological implausibilities. A notable example is the weight transport problem, which posits that the synaptic weight responsible for forwarding information is identical to that which conveys error information backward, as highlighted by Lillicrap et al. (2016). This issue has garnered substantial attention, contributing to the evolution of deep learning models that achieve performance levels on par with backpropagation in large-scale image classification tasks, as discussed in (Xiao et al., 2018). Within predictive coding research, it has been shown that the removal of these features does not markedly impair classification performance (Millidge et al., 2020b). While it is important to emphasize that our proposed algorithm remains functional in the weight transport framework described in the aforementioned study, testing it goes beyond the scope of the paper. To conclude, other works have introduced predictive coding-like algorithms that do not rely on weight transport (Ororbia et al., 2022; Ororbia & Kifer, 2022; Ororbia et al., 2020).

## B  Pseudocodes of Z-IL and PC

---

**Algorithm 2** Learning a dataset $\mathcal{D} = y_i$ with PC.

---

1: **Require:** For every $i$, $x_i^{(0)}$ is fixed to $y_i$,
2: **for** $t = 0$ to $T$ **do**
3:     For every $i$ and $l$, update $x^{(l)}$ to minimize $F$ via Eq. (6)
4:     **if** t = T **then**
5:         For every $l$ update each $\theta^{(l)}$ to minimize $F$ via Eq. (7),
6:     **end if**
7: **end for**

---

---

**Algorithm 3** Learning one training pair $(s^{\text{in}}, s^{\text{out}})$ with Z-IL

---

1: **Require:** $x_0^L$ is fixed to $s^{\text{in}}$, $x_0^0$ is fixed to $s^{\text{out}}$.
2: **Require:** $x^{(l)} = \mu^{(l)}$ for $l \in \{1, ..., L-1\}$, and $t = 0$.
3: **for** $t = 0$ to $T$ **do**
4:     **for** each level $l$ **do**
5:         Update $x^{(l)}$ to minimize $F$ via Eq. (6)
6:     **end for**
7:     **if** $t = l$ **then**
8:         Update $\theta^{(l)}$ to minimize $F$ via Eq. (7).
9:     **end if**
10: **end for**

---

Table 4: Theoretical Efficiency of PC, Z-IL, BP, and iPC.

|  | One inference step | PC | Z-IL | BP | iPC |
|---|---|---|---|---|---|
| Number of MMs per weight update | $(2L-1)$ | $(2L-1)T$ | $(2L-1)(L-1)$ | $(2L-1)$ | $(2L-1)$ |
| Number of SMMs per weight update | 2 | $2T$ | $2(L-1)$ | $(2L-1)$ | 2 |

## C   ON THE EFFICIENCY OF PC, BP, AND IPC

In this section, we discuss the time complexity and efficiency of PC, BP, Z-IL, and iPC. We now start with the first three, and introduce a metric that we use to compute such complexity. This metric is the number of *simultaneous matrix multiplications* (*SMMs*), i.e., the number of non-parallelizable matrix multiplications needed to perform a single weight update. It is a reasonable approximation of running time, as multiplications are by far the most complex operation ($\approx \mathcal{O}(N^3)$) performed by the algorithm.

### C.1   COMPLEXITY OF PC, BP, AND Z-IL

**Serial Complexity:** To complete a single update of all weights, PC and Z-IL run for $T$ and $(L-1)$ inference steps, respectively. To study the complexity of the inference steps, we consider the number of *matrix multiplications* (*MMs*) required for each algorithm: One inference step requires $(2L-1)$ MMs: $L$ for updating all the errors, and $(L-1)$ for updating all the value nodes (Eq. (6)). Thus, to complete one weight update, PC and Z-IL require $(2L-1)T$ and $(2L-1)(L-1)$ MMs, respectively. Note also that BP requires $(2L-1)$ MMs to complete a single weight update: $L$ for the forward, and $(L-1)$ for the backward pass. These numbers are summarized in the first row of Table 4. According to this measure, BP is the most efficient algorithm, Z-IL ranks second, and PC third, as in practice, $T$ is much larger than $L$. However, this measure only considers the total number of matrix multiplications needed, without considering whether some of them can be performed in parallel, which could significantly reduce the time complexity. We now address this problem.

**Parallel complexity:** The MMs performed during inference can be parallelized across layers. In fact, computations in Eq. (6) are layer-wise independent, thus $L$ MMs that update all the error nodes take the time of only one MM if properly parallelized. Similarly, in Eq. (6), $(L-1)$ MMs that update all the value nodes take the time of only one MM if properly parallelized. As a result, one inference step only takes the time of 2 MMs if properly parallelized (since, as stated, it consists of updating all errors and values via Eq. (6)). Thus, one inference step takes 2 SMMs; one weight update with PC and Z-IL takes $2T$ and $2(L-1)$ SMMs, respectively. Since no MM can be parallelized in BP (the forward pass in the network and the backward pass of error are both layer-dependent), before performing a single weight update, $(2L-1)$ SMMs are required. These numbers are summarized in the second row of Table 4. Overall, measured over SMMs, BP and Z-IL are equally efficient (up to a constant factor), and faster than PC.

### C.2   COMPLEXITY OF IPC

To complete one weight update, iPC requires one inference step, thus $(2L-1)$ MMs or 2 SMMs, as also demonstrated in the last column of Table 4. Compared to BP, iPC takes around $L$ times less SMMs per weight update, and should hence be significantly faster in deep networks. Intuitively, this is because matrix multiplications in BP have to be done sequentially along layers, while the ones in iPC can all be done in parallel across layers (Fig. 5). More formally, we have the following theorem, which holds when performing full-batch training:

**Theorem 3.1.** *Let $M$ and $M'$ be two equivalent networks with $L$ layers trained on the same dataset. Let $M$ (resp., $M'$) be trained using BP (resp., iPC). Then, the time complexity measured by SMMs needed to perform one full update of the weights is $\mathcal{O}(1)$ and $\mathcal{O}(L)$ for iPC and BP, respectively.*

*Proof.* Consider training on an MLP with $L$ layers, and update weights for multiple times on a single data point. Generalizations to multiple data points and multiple mini-batches are similar and will be provided after. We first write the equations needed to be computed for iPC to produce one weight

update:

$$x_{i,t}^{(L)} = s_i^{in} \text{ and } x_{i,t}^{(0)} = s_i^{out}$$

$$\hat{x}_{i,t}^{(l)} = \sum_{j=1}^{n^{l-1}} \theta_{i,j}^{(l+1)} f(x_{j,t}^{(l+1)}) \qquad \text{for } l \in \{1, \dots, L\} \qquad (9)$$

$$\varepsilon_{i,t}^{(l)} = x_{i,t}^{(l)} - \hat{x}_{i,t}^{(l)} \qquad \text{for } l \in \{1, \dots, L\}$$

$$x_{i,t+1}^{(l)} = x_{i,t}^{(l)} + \gamma \cdot \left( -\varepsilon_{i,t}^{(l)} + f'(x_{i,t}^{(l)}) \sum_{k=1}^{n^{(l+1)}} \varepsilon_{k,t}^{(l+1)} \theta_{k,i}^{(l)} \right) \quad \text{for } l \in \{1, \dots, L\} \qquad (10)$$

$$\theta_{i,j,t+1}^{(l)} = \theta_{i,j,t}^{(l)} - \alpha \cdot \varepsilon_{i,t}^{(l+1)} f(x_{j,t}^{(l)}) \quad \text{for } l \in \{1, \dots, L\}. \qquad (11)$$

We then write the three equations needed to be computed for BP to produce one weight update:

$$x_{i,t}^0 = s_i^{in}$$

$$x_{i,t}^{(l)} = \sum_{j=1}^{n^{l-1}} \theta_{i,j}^{(l+1)} f(x_{j,t}^{(l+1)}) \text{ for } l \in \{1, \dots, L\} \qquad (12)$$

$$\varepsilon_{i,t}^{(L)} = s_i^{out} - x_{i,t}^{(L)}$$

$$\varepsilon_{i,t}^{(l)} = f'\left(x_{i,t}^{(l)}\right) \sum_{k=1}^{n^{(l+1)}} \varepsilon_{k,t}^{(l+1)} \theta_{k,i}^{(l)} \text{ for } l \in \{L, \dots, 1\} \qquad (13)$$

$$\theta_{i,j,t+1}^{(l)} = \theta_{i,j,t}^{(l)} - \alpha \cdot \varepsilon_{i,t}^{(l+1)} f(x_{j,t}^{(l)}) \text{ for } l \in \{1, \dots, L\}.$$

First, we notice that the matrix multiplication (MM) is the most complex operation. Specifically, for two adjacent layers with the size of $n^l$ and $n^{l+1}$, the complexity of MM is $\mathcal{O}(n^l n^{l+1})$, but the maximal complexity of the other operations is $\mathcal{O}(\max(n^l, n^{l+1}))$. In the above equations, only equations with MM are numbered, which are the equations that we investigate in our complexity analysis.

Eq. (9) for iPC takes $L$ MMs, but one SMM, since the for-loop for $l \in \{1, \dots, L\}$ can run in parallel for different $l$. This is further because the variables on the right side of Eq. (9) are immediately available. Differently, Eq. (12) for iPC takes $L$ MMs, and also $L$ SMMs, since the for-loop for $l \in \{1, \dots, L\}$ has to be executed one after another, following the specified order $\{2, \dots, L\}$. This is further because the values on the right side of Eq. (12) are immediately available, but require to solve Eq. (12) again for another layer. That is, to get $x_{i,t}^{(L)}$, Eq. (12) has to be solved recursively from $l = 1$ to $l = L$.

Similar sense applies to the comparison between Eqs. (10) and (13). Eq. (10) for iPC takes $L-1$ MMs but 1 SMMs; Eq. (13) for BP takes $L-1$ MMs and also $L-1$ SMMs.

Overall, Eqs. (9) and (10) for iPC take $2L-1$ MMs but 2 SMMs; Eqs. (12) and (13) for BP take $2L-1$ MMs and also $2L-1$ SMMs. Then, the time complexity measured by SMMs needed to perform one full update of the weights is $\mathcal{O}(1)$ and $\mathcal{O}(L)$ for iPC and BP, respectively.

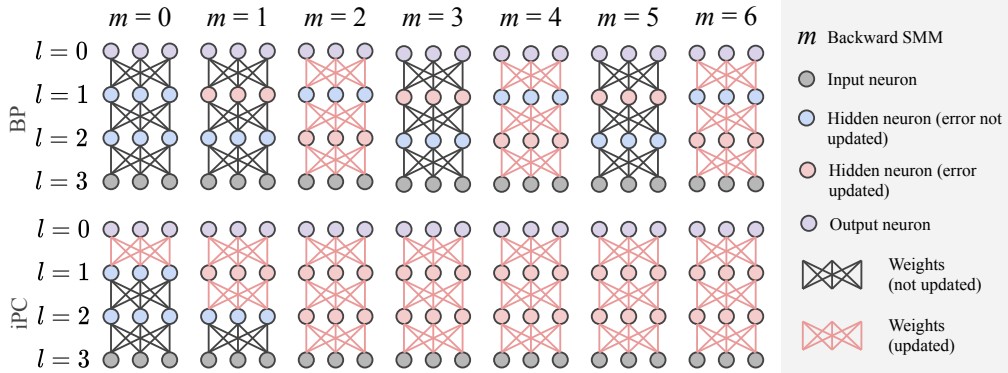

Figure 5: Graphical illustration of the efficiency over backward SMMs of BP and iPC on a 3-layer network. iPC never clears the error (red neurons), while BP clears it after every update. This allows iPC to perform 5 full and 2 partial updates of the weights in the first 6 SMMs. In the same time frame, BP only performs 3 full updates. Note that the SMMs of forward passes are excluded for simplicity, w.l.o.g., as the insight from this example generalizes to the SMMs of the forward pass.

## C.3 Efficiency on One Data Point

To make the difference more visible and provide more insights, we explain this in detail with a sketch of this process on a small network in Fig. 5, where the horizontal axis of $m$ is the time step measured by simultaneous matrix multiplications (SMMs), i.e., within a single $m$, one can perform one matrix multiplication or multiple ones in parallel; if two matrix multiplications have to be executed in order (e.g., the second needs results from the first), they will need to be put into two steps of $m$. Note that we only consider the matrix multiplications for the backward pass, i.e., the matrix multiplications that backpropagate the error of a layer from an adjacent layer for BP and the inference of Eq. (6) for iPC, thus the horizontal axis $m$ is strictly speaking "Backward SMM". The insight for the forward pass is similar as that of the backward pass. As mentioned above, for BP, backpropagating the error from one layer to an adjacent layer requires one matrix multiplication; for iPC, one step of inference on one layer via Eq. (6) requires one matrix multiplication. BP and iPC are presented in the first and second rows, respectively. Before both methods are able to update weights in all layers, they need two matrix multiplications for spreading the error through the network, i.e., a weights update of all layers occurs for the first time at $m = 2$ for both methods. After $m = 2$, BP cleared all errors on all neurons, so at $m = 3$, BP backpropagates the error from $l = 0$ to $l = 1$, and at $m = 4$, BP backpropagates the error from $l = 1$ to $l = 2$ after which it can make an update of weights at all layers again for the second time. Note that the matrix multiplication that backpropagates errors from $l = 1$ to $l = 2$ at $m = 4$ cannot be put at $m = 3$, as it requires the results of the matrix multiplication at $m = 3$, i.e., it requires the error to be backpropagated to $l = 1$ from $l = 0$ at $m = 3$. However, this is different for iPC. After $m = 2$, iPC does not reset $x_{i,t}^l$ to $\mu_{i,t}^l$, i.e., the error signals are still held in $\varepsilon_{i,t}^l$. At $m = 3$, iPC performs two matrix multiplications in parallel, corresponding to two inferences steps at two layers, $l = 1$ and $l = 2$, updating $x_{i,t}^l$, and hence the error signals are held in $\varepsilon_{i,t}^l$ of these two layers. Note that the above two matrix multiplications of two inference steps can run in parallel and be put into a single $m$, as inference requires only locally and immediately available information. In this way, a weight update in iPC is able to be performed at every $m$ ever since the very first few steps of $m$.

## C.4 CPU IMPLEMENTATION

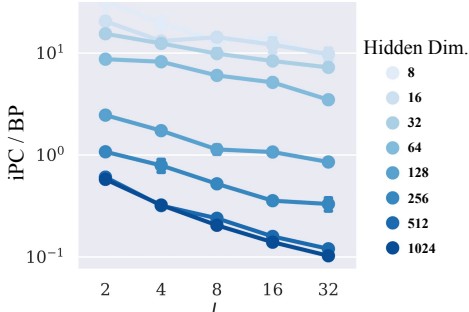

Figure 6: Ratio of the actual running time needed to perform a single weight update between BP and iPC on a CPU. Every dot represents a model; if the model lies below the horizontal line with label $10^0$, its weight update performed using iPC is faster than one performed using BP.

To further provide evidence of the efficiency of iPC with respect to BP, we have implemented the parallelization of iPC on a CPU, and compared it to BP, also implemented on CPU. We compute the time in milliseconds (ms) needed to perform one weight update of both on a randomly generated data point. In Fig. 6, we have plotted the ratio

*ms of iPC / ms of BP*

for architectures with different depths and widths. The results show that our naive implementation adds a computational overhead given by communication and synchronization across threads that makes iPC slower than BP on small architectures (hidden dimension $\leq 64$). However, this difference is inverted in large networks: in the most extreme case, one weight update on a network with 32 hidden layers and 1024 parameters per layer using iPC is 10 times faster than that using BP. This is still below the result of Theorem 3.1 due to the large overhead introduced in our implementation.

## D TRAINING DETAILS

We now list some additional details to reproduce our results.

### D.1 EXPERIMENTS OF EFFICIENCY

The experiments for the efficiency of generative models were run on fully connected networks with 128, 256, or 512 hidden neurons, and $L \in \{4, 5\}$. Every network was trained on CIFAR10 or Tiny ImageNet with learning rates $\alpha = 0.00005$ and $\gamma = 0.5$, and $T \in \{8, 12, 16\}$. The experiments on discriminative models are performed using networks with 64 hidden neurons, depth $L \in \{3, 4, 6\}$, and learning rates $\alpha = 0.0001$ and $\gamma = 0.5$. The networks trained with BP have the same learning rate $\alpha$. All the plots for every combination of hyperparameters can be found in Figs. 8 and 7.

### D.2 EXPERIMENTS OF GENERALIZATION QUALITY

As already stated in the paper body, to make sure that our results are not the consequence of a specific choice of hyperparameters, we performed a comprehensive grid search on hyperparameters, and reported the highest accuracy obtained, and the search is further made robust by averaging over 5 seeds. Particularly, we tested over 8 learning rates (from $0.000001$ to $0.01$), 4 values of weight decay $(0.0001, 0.001, 0.01, 0.1)$, and 3 values of the integration step $\gamma$ $(0.1, 0.5, 1.0)$. We additionally verified that the optimized value of each hyperparameter lies within the searched range of that hyperparameter. As for additional details, we used standard PyTorch initialization for the parameters. For the hardware, we used a single Nvidia GeForce RTX 2080 GPU on an internal cluster. Despite the large search, most of of the best results were obtained using the following hyperparameters: $\gamma = 0.5$ ($\gamma = 1$ for AlexNet), $\alpha = 0.00005$.

In the experiments on AlexNet, under all the hyperparameter combinations tested, iPC only failed to converge in some cases where the learning rate of the weights was the largest (0.01). In total, it converged 88 times out of 96 combinations of hyperparameters. This is not the case for PC, which converged in only 26 combinations of hyperparameters out of 96.

### D.3 LANGUAGE MODELS: IMPLEMENTATION DETAILS

For PC and iPC, we use the $\mathcal{F}_{KL}$ modification (Pinchetti et al., 2022), which replaces the Gaussian distribution for the layers with *softmax* activation with a non-Gaussian distribution. Initial experiments confirm that this helps both PC and iPC for both language models. Initial experiments with the masked language models also suggest that PC and iPC benefit from disabling the layer normalization (Ba et al., 2016) found in BERT, but BP does not. Therefore, for the masked language model, we assume layer normalization for BP only. For the conditional language models, we observe that iPC and BP benefit from layer normalization, but PC does not.

#### D.3.1 HYPERPARAMETERS

We train each model on 16 epochs of the training dataset. We do early stopping every $100 *$ $128/batch\_size$ batches. For iPC, and PC, we have observed that a batch size of 128 works best, so we assume it throughout the experiments.

Based on early experiments, we assume the parameter (weight) optimizer to be AdamW (Loshchilov & Hutter, 2017), and the inference optimizer to be stochastic gradient descent (SGD). We denote the parameter (weight) learning rate as $lr$, and the inference learning rate as $x\_lr$.

For PC, we also use an inference learning rate discount (between 0 and 1), which we use to multiply the learning rate by if the energy has not improved. We have observed that this can increase the performance and decrease the optimal $T$ (the number of updates per batch), hence speeding up the training. We use 0.9 for the discount value, which we have found works best. For iPC, we assume the $x\_lr$ value to be 0.5, as suggested by our initial experiments. We do grid search over the following hyperparameter ranges:

Masked language model:

- **BP:** $batch\_size \in \{32, 64, 128, 256, 512\}$,
  $lr \in \{0.0002, 0.0004, 0.0008, 0.0016, 0.0032, 0.0064, 0.0128\}$.
  Best combination: $batch\_size = 256$, $lr = 0.0032$.
- **PC:** $T \in \{5, 6, 7, 8, 9, 10, 11, 12\}$,
  $x\_lr \in \{0.015625, 0.03125, 0.0625\}$,
  $lr \in \{0.0001, 0.0002, 0.0004, 0.0008\}$.
  Best combination: $T = 8$, $x\_lr = 0.03125$, $lr = 0.0004$.
- **iPC:** $T \in \{4, 5, 6, 7, 8\}$,
  $lr \in \{0.0004, 0.0008, 0.0016, 0.0032\}$.
  Best combination: $T = 6$, $lr = 0.0008$.

Conditional language model:

- **BP:** $batch\_size \in \{32, 64, 128, 256, 512\}$,
  $lr \in \{0.0002, 0.0004, 0.0008, 0.0016, 0.0032, 0.0064, 0.0128\}$.
  Best combination: $batch\_size = 64$, $lr = 0.0016$.
- **PC:** $T \in \{7, 8, 9, 10, 11, 12\}$,
  $x\_lr \in \{0.00390625, 0.0078125, 0.015625, 0.03125\}$,
  $lr \in \{0.0001, 0.0002, 0.0004, 0.0008\}$.
  Best combination: $T = 10$, $x\_lr = 0.0078125$, $lr = 0.0004$.
- **iPC:** $T \in \{3, 4, 5, 6\}$,
  $lr \in \{0.0008, 0.0016, 0.0032, 0.0064, 0.0128\}$.
  Best combination: $T = 4$, $lr = 0.0032$.

### D.3.2 PER-SEED RESULTS

Conditional LM results:

PC: [201.096, 206.322, 198.798, 187.0, 214.385, 213.66, 218.577, 221.076, 219.609, 182.898]

iPC: [137.086, 138.435, 235.153, 160.08, 139.284, 140.68, 149.148, 237.234, 281.977, 133.098]

BP: [112.655, 113.454, 113.454, 113.235, 113.726, 115.153, 112.391, 112.877, 113.359, 112.921]

Masked LM results:

PC: [503.548, 1042.125, 1158.864, 909.209, 557.766, 1155.093, 1176.144, 901.295, 507.916, 1162.502]

iPC: [93.998, 93.899, 93.449, 137.027, 94.491, 108.241, 92.411, 97.724, 156.112, 94.504]

BP: [83.619, 147.775, 787.334, 133.073, 358.486, 455.535, 92.951, 95.322, 152.135, 135.3]

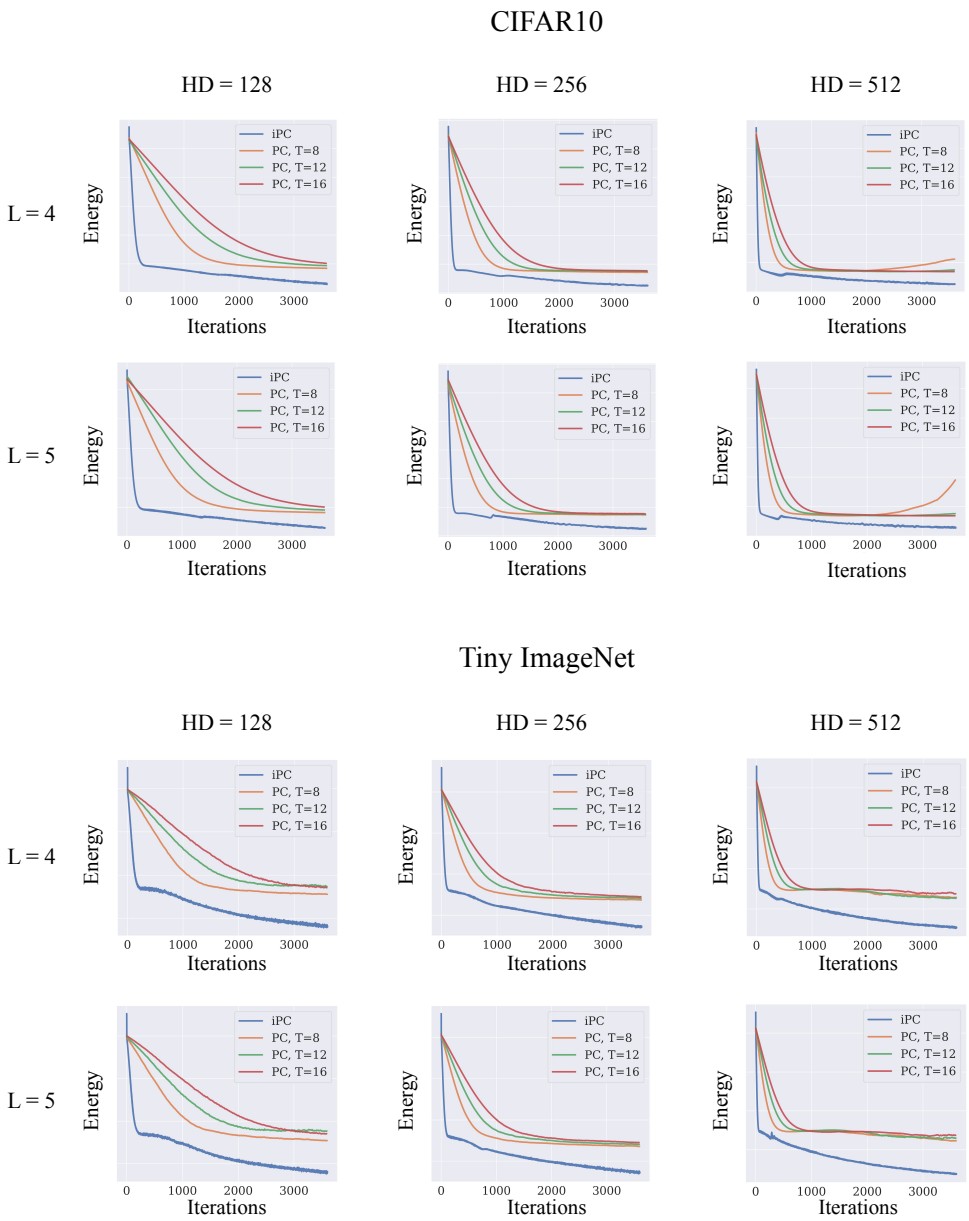

Figure 7: Decrease of the energy of generative models as a function of the number of iterations performed from the beginning of the training process. These experiments follow the same procedure of the ones in Fig. 2 (left), and serve the purpose of showing that the results that we provided are solid and robust under changes of hyperparameters.

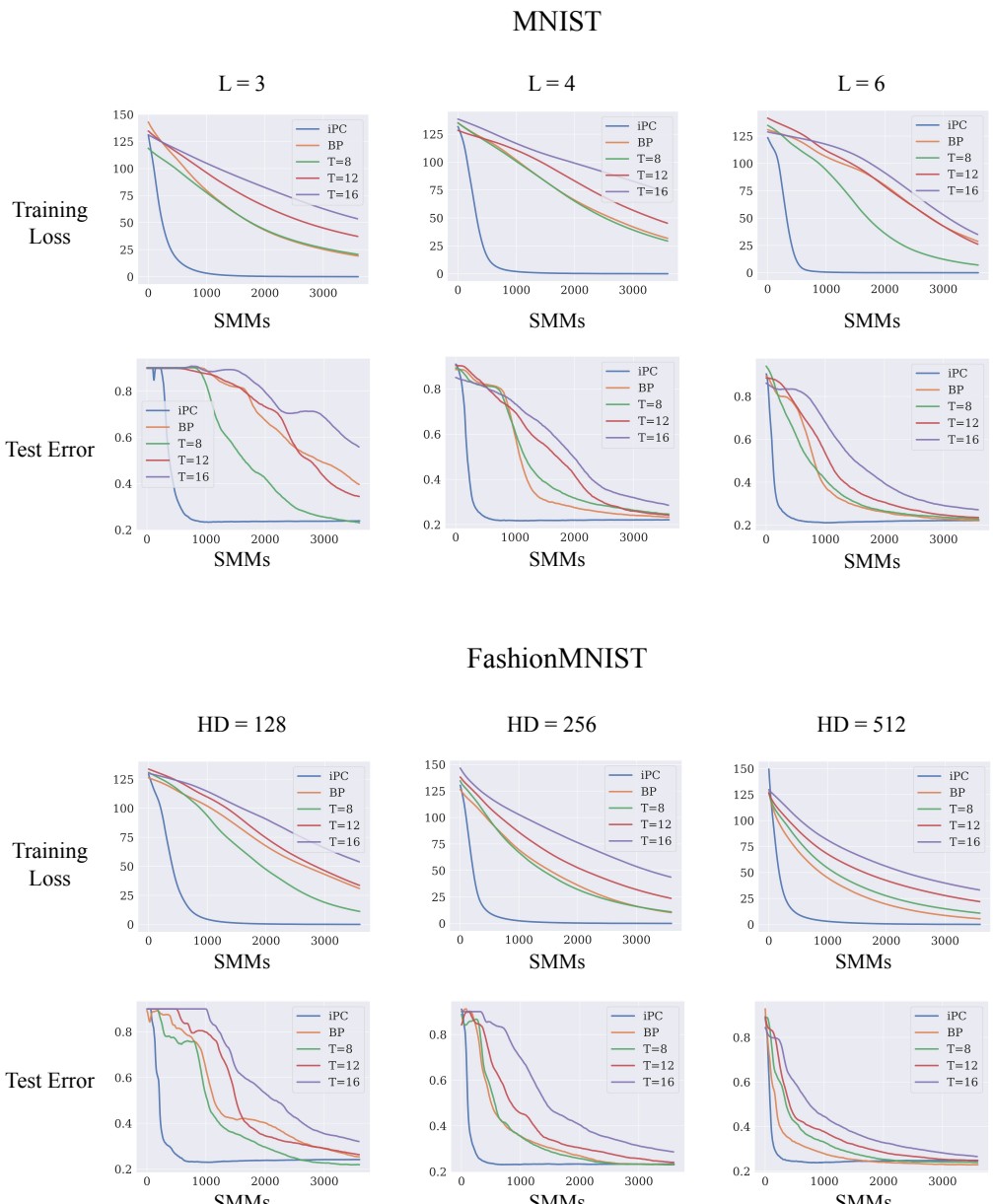

Figure 8: Training loss and test errors of different classifiers in a full-batch training regime as a function of the number of non-parallel matrix multiplications performed from the beginning of the training process. These experiments follow the same procedure of the ones in Fig. 2 (right), and serve the purpose of showing that the results that we provided are solid and robust under changes of hyperparameters.

# E ROBUSTNESS

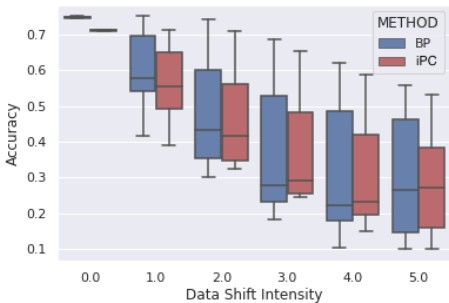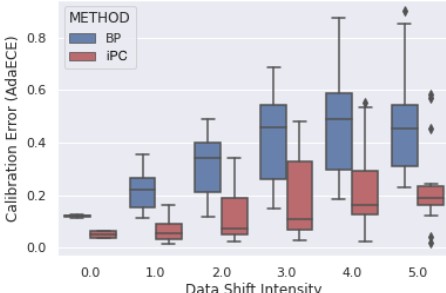

Figure 9: Larger image of the one provided in the main body of this work. Robustness of BP and iPC under distribution shift (AlexNet on CIFAR10 under five different intensities of the corruptions rotation, Gaussian blur, Gaussian noise, hue, brightness, and contrast). *Left:* Comparable decline of model accuracy between BP and iPC. *Right:* iPC maintains model calibration significantly better than BP under distribution shift.

