# OpenReview forum: "A Stable, Fast, and Fully Automatic Learning Algorithm for Predictive Coding Networks"
_ICLR.cc/2024/Conference — ICLR 2024 poster_

### Official Review · Reviewer_seiB · 2023-10-22

**Soundness:** 3 good
**Presentation:** 3 good
**Contribution:** 2 fair
**Rating:** 3
**Confidence:** 4

**Summary:**

The paper proposes incremental predictive coding. At a high level, predictive coding is the idea that one way to see brain function is that there is a top down generative model whose parameters are updated based on discrepancy between model output and stimulus received from the environment. More specifically, this paper talks about a specific generative model: a hierarchical Gaussian generative model where the means are passed thru non-linear functions from layer to layer of a generative neural network, with the covariance held fixed at identity. The optimal solution to the generative model (given data) is of course intractable and the next best is the standard expectation maximization update algorithm. The authors point out that this is generally very slow since the expectation step takes a very long time to converge when you have multiple layers in the neural network/hierarchical model. The authors propose to overcome the speed issue with incremental expectation maximization ala Neal and Hinton where gradient steps are taken with the expectation and maximization step either interleaved or in parallel.

**Strengths:**

The paper does an excellent job of introducing the topic of predictive coding. For the particular case of a hierarchical Gaussian generative model, (which is the generative model this paper is concerned with), the paper meticulously works out the details of the EM objective function. And finally, the paper has comprehensive experiments demonstrating that incremental EM does better than EM timewise, when implemented for the neural network modeling the h-Gaussian generative model.

**Weaknesses:**

Unfortunately, all of the results are derivative of Neal and Hinton's incremental EM where they showed that both the expectation and the maximization step can be done at the same time and is a theoretically sound way of doing decent for model parameters. There is nothing that this paper contributes conceptually beyond this well known result. The hierarchical Gaussian generative model is also very well known. All in all, the paper lacks sorely in novelty of contribution.

**Questions:**

None

---

> ### Author Response · Authors · 2023-11-15
> **Rebuttal**
>
> > Unfortunately, all of the results are derivative of Neal and Hinton's incremental EM where they showed that both the expectation and the maximization step can be done at the same time and is a theoretically sound way of doing decent for model parameters. There is nothing that this paper contributes conceptually beyond this well known result. The hierarchical Gaussian generative model is also very well known. All in all, the paper lacks sorely in novelty of contribution.
>
> We agree that iPC is mainly obtained by applying incremental EM to the original formulation of PC, and it is hence a mix of two existing techniques. We believe, however, that this should not be a reason to completely dismiss a research project, without first considering the impact that it could potentially have in the relevant literature. Let us highlight the reasons why we believe this work could be impactful in this field:
>
> Going back in time, the first work introducing the weight update of PC is Rao and Ballard’s in 1999. From Rao and Ballard’s paper to today, countless works use the original rule to update the weights, which is much slower than ours, and less performant in the tasks that we have proposed. Researchers performing experiments have certainly experienced and struggled with the bad efficiency and stability of PC models. Figuring out a new update rule for the weights that addresses this problem, without altering bio-plausibility or adding additional drawbacks, is a problem of vital importance in the field. Our proposed iPC hence addresses an existing practical problem that a large community has had for years. We foresee that many colleagues working in PC or similar fields could now use this update rule, instead of the standard one, as it results in a big gain of efficiency, as well as theoretical guarantees in terms of convergence, with (according to our research) no drawbacks.
>
> The fact that it is much more stable than the original formulation of PC, and constantly outperforms it by converging to better minima is not a negligible detail, as it allows PC to reach a performance comparable to those of BP on complex tasks, such as training deep convolutional models, or language models. Such results are novel in computational neuroscience. This is important, as it allows researchers to improve the performance of their experiments. We have given evidence that this is the case via a large number of experiments: generative models trained on complex datasets such as CIFAR10 and Tiny ImageNet, and discriminative models of different kinds (convolutional and feedforward) on a large number of datasets.
>
> We also believe that our work has the potential to impact research in computational neuroscience, as it does not alter the biological plausibility of the original formulation of PC in any way. Actually, the lack of requirement for a control signal that updates the weights makes the model even more plausible biologically. Hence, this will allow our algorithm to be used in theoretical research in computational neuroscience, where the goal is to simulate brain behaviors and not improve the performance of machine learning models. All in all, we believe that our contribution has the potential to be impactful enough in the community of neuroscience-inspired learning, regardless of the simplicity of its update rule.

---

> > ### Comment · Reviewer_seiB · 2023-11-17
> >
> > The authors agree with my assessment: And I quote "We agree that iPC is mainly obtained by applying incremental EM to the original formulation of PC, and it is hence a mix of two existing techniques. "

---

### Official Review · Reviewer_sVBr · 2023-10-28

**Soundness:** 3 good
**Presentation:** 2 fair
**Contribution:** 2 fair
**Rating:** 6
**Confidence:** 4

**Summary:**

The paper introduces the incremental Predictive Coding (iPC) framework, a more biologically-plausible version of PC networks. It addresses the update locking problem in standard PC algorithms by parallelizing neural dynamics and learning steps using Neal \& Hinton's incremental expectation maximization. This advancement eliminates the need for a global control signal to switch between inference and learning steps. The paper evaluates the proposed method on image classification tasks and language models, comparing it to standard PC and backpropagation (BP) algorithms.

**Strengths:**

* The paper addresses an important problem in the field of biologically-plausible learning and is generally well-written.
* The efficiency gained by iPC is a notable contribution.
* The numerical experiments showcase improved results compared to standard PC and comparable results to BP.
* The experimental descriptions in the appendix and code availability enhance reproducibility.

**Weaknesses:**

The paper has some clarity issues in my opinion. Please see the following items and the questions section.

* Regarding the following sentence in the page 3: "For a detailed formulation on how this energy function is derived from the variational free energy of Eq. 2, we refer to ..., or to **the supplementary**", which supplementary section includes this derivation? I do not see it.

* The caption of Table 1 should specify whether these are train or test accuracies.

* The sentence preceding the "Comparison with BP" section in page 6 seems redundant (seems to repeat the previous sentence); consider removing it if not necessary.

* Could you clarify which dataset the statement "reported the highest accuracy obtained" in Figure 3 refers to (train or test set)?

* The statement about iPC performing better on standard CNN than on AlexNet contradicts Table 1. According to Table 1, CNN accuracy is around 72\% whereas AlexNet accuracy is around 80\%. Therefore, iPC does not perform better on standard CNN than on AlexNet.

* The proposed method has one aspect that is not bio-plausible: it suffers from the weight transport problem, i.e. Eq. 6 requires the transpose of the forward mapping. I think this limitation can be discussed in Appendix section A (A Discussion on Biological Plausibility).

* There is an artificial empty space in the top right of page 19 caused by the caption of Figure 6.

* I think the plots in Figure 7 should include y-axis labels. Are they accuracy? energy? loss?

* I think the captions of Figure 7 and 8 should include more explanations.

**Questions:**

* It is not really intuitive  to me why PC with smaller T values converges faster (in terms of Energy in Figure 2). Could you give an insight about it, or elaborate it?

* What is the meaning of the notation $\mathcal{O}(\text{max} n^l, n^l)$ on page 17?

**Details Of Ethics Concerns:**

No ethical concerns were identified in this paper.

---

> ### Author Response · Authors · 2023-11-15
>
> We thank the reviewer for the thorough read, and for all the pointers, spotted typos, and suggestions. They have all been addressed in the updated manuscript. We believe that this definitely improves the clarity of the manuscript. More generally, every time that we discuss accuracies, they are test accuracies (now addressed everywhere in the text).
>
> About the weight transport, we have added the following, where pointed out by the reviewer:
>
> Weight Transport While predictive coding is designed to model information processing in various brain regions, the current formulation still exhibits certain biological implausibilities. A notable example is the weight transport problem, which posits that the synaptic weight responsible for forwarding information is identical to that which conveys error information backward, as highlighted by Lillicrap et al, 2016. This issue has garnered substantial attention, contributing to the evolution of deep learning models that achieve performance levels on par with backpropagation in large-scale image classification tasks, as discussed in Xiao et al. 2019. Within predictive coding research, it has been shown that the removal of these features does not markedly impair classification performance highlighted in Millidge et al.2020. While it is important to emphasize that our proposed algorithm remains functional in the weight transport framework described in the aforementioned study, testing it goes beyond the scope of the paper. To conclude, other works have introduced predictive coding-like algorithms that do not rely on weight transport (Ororbia & Kiefer, 2022).
>
> Questions:
>
> > why PC with smaller T values converges faster (in terms of Energy in Figure 2). Could you give an insight about it, or elaborate it?
>
> We believe that the reason is that out-of-equilibrium updates are be better (and of larger magnitude) than updates performed to models closer to convergence. This has been shown to be consistent in a wide range of experiments.
>
> > What is the meaning of the notation on page 17?
>
> Thank you again, this was a typo: the correct notation is O(max(n^l,n^{l+1}). This has been updated in the manuscript.

---

> > ### Comment · Reviewer_sVBr · 2023-11-18
> >
> > Dear authors,
> >
> > Thank you very much for your answer. I would like to bring to your attention that two of my questions remain unanswered:
> > > The statement about iPC performing better on standard CNN than on AlexNet contradicts Table 1. According to Table 1, CNN accuracy is around 72\% whereas AlexNet accuracy is around 80\%. Therefore, iPC does not perform better on standard CNN than on AlexNet.
> >
> > > Regarding the following sentence in the page 3: "For a detailed formulation on how this energy function is derived from the variational free energy of Eq. 2, we refer to ..., or to the supplementary", which supplementary section includes this derivation? I do not see it.
> >
> > Therefore, I would like to kindly ask again:
> > * 1) Is there any possibility of a typographical error in Table 1?
> > * 2) Have I overlooked the supplementary section that elucidates the derivation of the energy function in Eq. (5) from the variational energy in Eq. (2)?
> >
> > Thanks in advance.

---

> > > ### Author Response · Authors · 2023-11-18
> > >
> > > Dear reviewer,
> > >
> > > Thank you for your reply. The two statements you pointed out have been removed in our previous update of the manuscript (we had already spotted them, right after submission), as they were residuals of a previous iteration of the manuscript.
> > >
> > > More in detail:
> > >
> > >
> > > > Is there any possibility of a typographical error in Table 1?
> > >
> > > There are no typographical errors in the tables. The now deleted sentence refers to experiments on Alexnet performed with a slightly different setup: the paper now reports the result by using exactly the original implementation, that reshapes the size of the images to 224x224. In a previous iteration, this was not the case.
> > >
> > > > Have I overlooked the supplementary section that elucidates the derivation of the energy function in Eq. (5) from the variational energy in Eq. (2)?
> > >
> > > No, you have not: the (now deleted) reference to the supplementary material refers to a section that we decided to remove before submission: the derivation of the energy function we proposed was similar to that of another work [1]. For a matter of correctness, we had decided to remove it, and simply refer the reader to the original work.
> > >
> > > [1] Millidge, Beren, Anil Seth, and Christopher L. Buckley. "Predictive coding: a theoretical and experimental review." arXiv preprint arXiv:2107.12979 (2021).

---

> > > > ### Comment · Reviewer_sVBr · 2023-11-18
> > > >
> > > > Dear authors,
> > > >
> > > > I would like to thank you again for your prompt response. I appreciate the clarification provided, and I think that all my questions have been addressed. I want to maintain my score.
> > > >
> > > > Thanks.

---

### Official Review · Reviewer_Tect · 2023-10-30

**Soundness:** 4 excellent
**Presentation:** 3 good
**Contribution:** 3 good
**Rating:** 6
**Confidence:** 4

**Summary:**

This paper proposes iterative predictive coding (iPC) a predictive coding algorithm aims to be faster and more scalable than traditional predictive coding approaches. The key algorithmic innovation is that both intermediate layer activations and weights are simultaneously minimized (well, alternating training every step) rather than allowing hidden state activations to equilibrate as in previous models. In a series of experiments using Gaussian models, iPC outperforms competing approaches.

**Strengths:**

- Advances the state of the art for predictive coding, which remains a highly promising idea without a commensurately performant implementation.
- Extends PC results beyond the small-scale experiments in previous work.
- Brings PC algorithms closer to biological reality, where computations run simultaneously in real time.
- Strong set of experiments against comparison models.

**Weaknesses:**

- I came away with the impression that the key algorithmic innovation is simply to alternate weight and hidden layer activations every step (without running either to convergence). It's quite hard to believe this hasn't been tried before. Is there something new here that makes this work?
- The analysis is limited to Gaussian networks, which limits the range of applicability of the learning rules in (6) and (7).
- While the experiments are good, there is not much insight provided as to *why* this new approach works better.

**Questions:**

- The analysis in the paper is focused on feedforward Gaussian networks, but can these ideas be extended to networks with substantial recurrence? In the original PC formulation, it's these recurrent inputs that carry error signals, but I'm curious as to whether the "out of equilibrium" method proposed here would also be able to dynamically balance multiple kinds of feedback.
- Put another way: how much biological plausibility is lost if the Gaussian and feedforward assumptions are relaxed to mirror more cortical-like networks.

---

> ### Author Response · Authors · 2023-11-15
> **Rebuttal**
>
> We thank the reviewer for the feedback and the time spent on the manuscript.
>
> > the key algorithmic innovation is simply to alternate weight and hidden layer activations every step (without running either to convergence). It's quite hard to believe this hasn't been tried before. Is there something new here that makes this work?
>
> This is completely novel and has never been tried before in the field of predictive coding networks (to our knowledge).
> It is important that you mentioned the alternation of the updates of weights and activations, as well as the fact that the algorithm does not run until convergence, as this is key to make iPC work: running iPC until convergence on every batch of data would result in a strong overfitting of the last batch presented, as well as a forgetting of the previously learned information. It is the combination of both that allows the model to generalise well on unseen data (as well as making it much more efficient).
> This is different from other works, which tend to run every iteration for a lot of time steps, or until convergence (for example, in the paper *Predictive coding approximates backprop along arbitrary computation graphs* the authors use T=200 [1]).
>
>
> > The analysis is limited to Gaussian networks, which limits the range of applicability of the learning rules in (6) and (7).
>
>
> Note that predictive coding is defined on Gaussian networks only: linear prediction errors arise naturally from this formulation (they are the derivatives (or scores) of the variational free energy of a generative model formed by a hierarchy of Gaussians). Having different probability distributions, such as categorical, would result in update rules that do not have a structure of neuron-synapses-neuron, as well as prediction errors.
>
> That being so, this assumption is sometimes relaxed, to allow the algorithm to work on more complex generative models. An example of that is the experiment on Transformer networks that we presented in the last section, where we have added categorical distributions after the attention mechanism, hence showing that iPC naturally generalises to non-Gaussian distributions as well.
>
>
> > While the experiments are good, there is not much insight provided as to why this new approach works better.
>
> Incremental updates mitigate the risk of overfitting by continuously adjusting the model parameters, preventing it from becoming too fixated on the specificities of the initial data that it was trained on. Standard predictive coding, on the other hand, tends to overfit on single data points. In terms of convergence guarantees, incremental methods are good in escaping local minima and reach more globally optimal solutions, in which non-incremental ones such as standard PC can get stuck. This is due to the continuous update of the parameters when the neural activities have not yet converged, making the dynamics more “noisy” [3].
>
> Questions:
>
> > The analysis in the paper is focused on feedforward Gaussian networks, but can these ideas be extended to networks with substantial recurrence? In the original PC formulation, it's these recurrent inputs that carry error signals, but I'm curious as to whether the "out of equilibrium" method proposed here would also be able to dynamically balance multiple kinds of feedback.
>
> It can naturally be extended to recurrent cases, such as [4]: note that the final update rule is strictly local. Hence, it is not aware of the structure of the model: they will work without any modifications in models with cycles, loop and recursions.
>
>
> >  how much biological plausibility is lost if the Gaussian and feedforward assumptions are relaxed to mirror more cortical-like networks.
>
>
> If by cortical-like networks you mean entangled and cyclic structures that resemble brain regions, our method can be applied with no modification (for examples of predictive coding models that use such structures, see [4]).
>
>
> [1] Millidge, Beren, Alexander Tschantz, and Christopher L. Buckley. "Predictive coding approximates backprop along arbitrary computation graphs." Neural Computation 34.6 (2022): 1329-1368.
>
> [2] Friston, Karl. "A theory of cortical responses." Philosophical transactions of the Royal Society B: Biological sciences 360.1456 (2005): 815-836.
>
> [3] Karimi, Belhal, et al. "On the global convergence of (fast) incremental expectation maximization methods." Advances in Neural Information Processing Systems 32 (2019).
>
> [4] Salvatori, Tommaso, et al. "Learning on arbitrary graph topologies via predictive coding." Advances in neural information processing systems 35 (2022): 38232-38244.

---

> > ### Comment · Reviewer_Tect · 2023-11-17
> >
> > I appreciate the authors' thorough responses to my questions. I believe adding some of these responses to a final draft would help illuminate the limitations and potential generalizations of the model.
> >
> > I also think the authors have been clear about the fact that this is an application of an existing idea to a new problem. I think there's value in that, particularly because 1) it's not been done in predictive coding; and 2) it substantially improves performance and brings these models closer to biological plausibility. The first point by itself isn't an argument for acceptance, but combined with the second, I think this represents a worthwhile contribution to the predictive coding and computational neuroscience literature.
> >
> > I plan to maintain my score at a 6 to reflect this mix but will raise my confidence to a 4.

---

### Official Review · Reviewer_Ctxj · 2023-11-01

**Soundness:** 3 good
**Presentation:** 3 good
**Contribution:** 3 good
**Rating:** 6
**Confidence:** 4

**Summary:**

Predictive coding presumably uses an EM-like algorithm for learning representation and for making inferences. The authors of this paper proposed a method called incremental predictive coding that allows all computations involving learning and inference to take place locally, simultaneously, and autonomously. The results speak for themselves. iPC achieves accuracy comparable to BP and even lower calibration error than BP.

**Strengths:**

It is rather impressive that iPC can achieve accuracy comparable to BP, and lower calibration error than BP.  If this unsupervised learning algorithm can run as fast as BP, it could be fairly significant.

**Weaknesses:**

Even though the work could be important and the paper is well-written, the authors have archived the paper in public,  revealing their identities. Hence, this review is no longer double-blinded. This is perhaps an unfortunate oversight.

**Questions:**

The intuition of WHY the algorithm works is not clearly explained.

**Details Of Ethics Concerns:**

A simple google search of the title of the paper indicates this paper has been archived in public as
https://arxiv.org/pdf/2212.00720.pdf
The paper describes the iPC algorithm, with same/similar writing, figures, and citations.
Obviously, this submission is no longer a double-blind review as they have certified.

---

> ### Author Response · Authors · 2023-11-11
> **Uploading research papers on ArXiv is allowed in the ICLR submission policy**
>
> Dear reviewer,
>
> First of all, we thank you for your time and for the positive comments and kind words about our work. Second, we would like to state that the policy submission of ICLR does allow for the ArXiv submission of our work, as highlighted in the rules, that we copy below:
>
> > **Dual Submission Policy**:
> Submissions that are identical (or substantially similar) to versions that have been previously published, or accepted for publication, or that have been submitted in parallel to this or other conferences or journals, are not allowed and violate our dual submission policy. However, papers that cite previous related work by the authors *and papers that have appeared on non-peer reviewed websites (like arXiv)* or that have been presented at workshops (i.e., venues that do not have publication proceedings) do not violate the policy. The policy is enforced during the whole reviewing process period. Submission of the paper to archival repositories such as arXiv is allowed during the review period.
>
> And again, in the double blind review:
>
> > **Double blind reviewing:**
> Submissions will be double blind: reviewers cannot see author names when conducting reviews, and authors cannot see reviewer names.  *Having papers on arxiv is allowed per the dual submission policy outlined below.*
>
> The above texts have been copied and pasted from the official “call for papers” webpage of ICLR 2024, that you can find here:  https://iclr.cc/Conferences/2024/CallForPapers .
>
> Hence, we did not violate any rule, as the preprint of our work on arXiv is allowed.
>
> To this end, would it be possible for you to raise the score in a way that reflects your true opinion about our work?
>
>  Many thanks,
>
> The authors.

---

> > ### Comment · Area_Chair_8Nr1 · 2023-11-17
> >
> > Dear reviewer Ctxj,
> >
> > Would you mind taking another look at the author rebuttal for this paper?
> >
> > For what it's worth, I believe they are correct that posting on arxiv does NOT constitute a violation of the double-blind review policy. (In fact, it is fairly common).  In light of that, would you revisit your assessment of the paper?
> >
> > Thanks,
> > AC

---

> > > ### Comment · Reviewer_Ctxj · 2023-11-17
> > > **Have increased the score**
> > >
> > > My earlier comments about double-bline is based on the line above
> > > "Anonymous Url: I certify that there is no URL (e.g., github page) that could be used to find authors' identity."
> > > Thank you for clarifying the specific Author instruction about permission to publish in arxiv. But in any case, it is really not double-blind, right?
> > >
> > > I do think the contribution is a good one and will have some impact in the field. Even though iPC being a marriage of two existing techniques, demonstrating a certain theoretical idea actually works in a specific important scenario is still important and meaningful. I would have given it a 6 or 7 had not been the double-blind issue. So I have now upgraded my score to a 6, since the only alternative would be an 8.

---

### Public Comment · ~Arjun_Kandaswamy1 · 2025-08-20
**what about at test time  algorithm**

how in test time the ,algorithm is used/modified  Algorithm 1

e^(0) how calculated as it contains labels , how it can be used in test time , for updating x^(1) latent.

can you give algorithm for testing.

---

### Meta-Review · Area_Chair_8Nr1 · 2023-12-12

**Metareview:**

This paper proposes a method for predictive coding called "incremental predictive coding", which allows all computations involving learning and inference to take place locally, simultaneously, and autonomously. One reviewer raised serious concerns about novelty, but the authors argued that -- even if the results rely on a mixture of existing techniques -- the quantitative improvements are impressive in their own right, that they provide theoretical guarantees in terms of convergence, and that the proposed framework represents a major improvement over the predominant methods employed in the field.  Three of the reviewers agreed with this argument, and I am inclined to side with the majority opinion as well. I believe the work will have a positive impact on the field and am pleased to see it presented at ICLR.  I would ask the authors to attend carefully to reviewer concerns about novelty, clarity, and motivation when preparing their final version.

**Justification For Why Not Higher Score:**

Reviewer seiB raised concerns about motivation, clarity, and (especially) novelty that left the paper quite close to the acceptance threshold, and the three more positive reviewers were unfortunately only willing to give it a just-above-threshold score. Thus it's not a strong candidate for a spotlight or oral presentation, in my opinion.

**Justification For Why Not Lower Score:**

Ultimately I thought the authors made a convincing case in their rebuttal that the paper would have a major impact (despite concerns about strict novelty of the ideas they used to come up with their method).  Thus it seems to me worth accepting, though I don't feel strongly (especially since none of the reviewers championed it strongly).

---

### Decision · Program_Chairs · 2024-01-16

Accept (poster)